

# Studying the impact of CI on pull request delivery time in open source projects—a conceptual replication

Yunfang Guo and Philipp Leitner

Software Engineering Division, Chalmers | University of Gothenburg, Gothenburg, Sweden

## ABSTRACT

Nowadays, continuous integration (CI) is indispensable in the software development process. A central promise of adopting CI is that new features or bug fixes can be delivered more quickly. A recent repository mining study by *Bernardo, da Costa & Kulesza (2018)* found that only about half of the investigated open source projects actually deliver pull requests (PR) faster after adopting CI, with small effect sizes. However, there are some concerns regarding the methodology used by Bernardo et al., which may potentially limit the trustworthiness of this finding. Particularly, they do not explicitly control for normal changes in the pull request delivery time during a project's lifetime (independently of CI introduction). Hence, in our work, we conduct a conceptual replication of this study. In a first step, we replicate their study results using the same subjects and methodology. In a second step, we address the same core research question using an adapted methodology. We use a different statistical method (regression discontinuity design, RDD) that is more robust towards the confounding factor of projects potentially getting faster in delivering PRs over time naturally, and we introduce a control group of comparable projects that never applied CI. Finally, we also evaluate the generalizability of the original findings on a set of new open source projects sampled using the same methodology. We find that the results of the study by Bernardo et al. largely hold in our replication. Using RDD, we do not find robust evidence of projects getting faster at delivering PRs without CI, and we similarly do not see a speed-up in our control group that never introduced CI. Further, results obtained from a newly mined set of projects are comparable to the original findings. In conclusion, we consider the replication successful.

## INTRODUCTION

Continuous Integration (CI) is by now a popular practice in the software community (*Duvall, Matyas & Glover, 2007*). CI helps developers integrate changes frequently in a collaborative manner. As a distributed and cooperative practice, CI is commonly used in both, commercial and open source software (OSS) development. Considerable previous research has investigated the impact of CI on OSS projects. *Vasilescu et al. (2015)* found that core developers are able to discover more bugs using CI. *Ståhl & Bosch (2014)* claim that integrators tend to release more frequently after adopting CI. Finally, a recent study

Corresponding author
Philipp Leitner,
philipp.leitner@chalmers.se

by *Bernardo, da Costa & Kulesza (2018)* empirically analyzed whether CI improves the time-to-delivery of merged Pull Requests (PRs) that are submitted to GitHub projects. Interestingly, this study revealed that only 51.3% of the analyzed OSS projects actually deliver merged PRs more quickly after adopting CI. The authors present an increase in PR submission numbers after adopting CI as a possible reason for this relatively counterintuitive result. Further, the authors used regression analysis to identify two factors (merge workload and queue rank) that are the main predictors of PR delivery time.

However, we observe that the study by Bernardo et al. exhibits some important limitations. Firstly, their methodology consists of comparing various PR related metrics before and after CI adoption without controlling for confounding factors, most importantly that PR delivery time may increase or decrease naturally over the lifetime of a project. For example, it is conceivable that projects may just naturally get better at merging PRs over time, independently of whether they adopt CI or not. Secondly, they do not make use of a control group of projects that never adopted CI in the first place. In our opinion, this limits the trustworthiness of the results of Bernardo et al.

Hence, in our work, we present a conceptual replication (*Shull et al., 2008*) of this study. We replicate their work and investigate the same research questions with slightly different methodology, and by incorporating additional study objects. Concretely, we investigate the following research questions.

**RQ1: Exact Replication.** *Can the original study results be reproduced?*

As a baseline, we reproduce the original results of the study, using the same methodology and the data provided by the authors. We are able to achieve very similar results, with minor differences (between 1.1 and 5.5 percentage points difference to the originally published results).

**RQ2: Conceptual Replication.**

To extend the original study methodology, and address the concerns we have with the experimental methodology as initially proposed, we investigate two different aspects:

*RQ2.1: Can similar results to the original study be found when controlling for changes in PR delivery time over the lifetime of a project?* To answer this question, we apply Regression Discontinuity Design –RDD (*Thistlethwaite & Campbell, 1960*), a statistical method that allowed us to evaluate whether there is a trend of PR delivery times over time, and whether this trend changes significantly when CI is introduced. We find no clear evidence of such trends in the data, alleviating our concerns in this regard. However, we observe that PR delivery times depend strongly on when in the release cycle a PR is merged. PRs that are merged close to the next release are released much quicker than PRs that come in shortly after a release. This indicates that, ultimately, CI introduction may have less impact on PR delivery times than how often a project releases.

*RQ2.2: Are there other factors besides merge workload and queue rank that strongly impact the PR delivery time?* Based on the results of RQ2.1, we hypothesize that one important factor impacting PR delivery time that is not directly captured in the original study is when in the release cycle a new PR is submitted. We incorporate this additional variable into the regression model, and evaluate whether it is a better predictor than the variables in the

original study. We find that this "come-in time" indeed is the best predictor of PR delivery time for a majority of projects.

**RQ3: Generalizability.**

Finally, to evaluate the generalizability of the results, we apply our adapted methodology to two new data sets, a new data set of study subjects collected using the same methodology as in the original study, and a control group of projects that have similar characteristics but have, to the best of our knowledge, never applied CI.

*RQ3.1: Can similar results be found when applying the same methodology to different projects that have also adopted CI?*

We find that results found for a new set of study subjects vary up to 14 percentage points. However, the high-level conclusions drawn by the original study still hold for our replication using new data. Hence, we consider the original findings to be largely confirmed using additional data, with the caveat that the individual differences between projects may be very high.

*RQ3.2: Can similar results be found when applying the same methodology to different projects that have never adopted CI?*

Finally, we collect a control group of comparable projects that have never adopted CI. We observe results that vary between 10 and 16 percentage points from what has been observed based in the original data, i.e., the results of applying the same methods on a control group are only mildly more different than applying the same method on a new test group (RQ3.1). However, we observe that projects in the control group do not increase the number of PRs they are able to handle per release over time. This is different to both test groups, where we observe a statistically significant increase in submitted, merged, and released PRs per release after CI adoption.

In summary, we consider the replication successful. Our concern regarding trends in the data has largely been alleviated, and an analysis of a control group has led to, at least subtly, different results. However, our results also indicate that PR delivery times seem to more strongly depend on when in the release cycle a PR comes in than on whether or not a CI system is present. This is consistent with the original study, which also reported that the presence of CI only impacts delivery time metrics with small effect sizes. Our study sheds some more light on why this is the case. Finally, we conclude that the delivery time of PRs is not strongly impacted by whether a project adopts CI, but projects that do are able to handle more PRs per release than projects that do not.

The present article is based on work conducted by the first author over five months in early 2018 as part of her master's thesis project at Chalmers University of Technology, under the supervision of the second author (*Guo, 2019*). The results presented here are a summary of this work, and more details can be found in the thesis report.

## BACKGROUND

We now present important background on CI and the pull request based development model. Further, we summarize the main results of *Bernardo, da Costa & Kulesza (2018)*, which we attempt to replicate in our study.

## CI and the pull request based development model

CI is a practice which has originated from Agile software development and Extreme Programming. Its core tenet is the merging of all developer working copies to shared mainline several times a day. Each integration is then verified by an automated build, which allows errors to be detected and located as early as possible (see also online https://www.thoughtworks.com/continuous-integration). CI promises manifold benefits, such as quickening the delivery of new functionalities (*Laukkanen, Paasivaara & Arvonen, 2015*), reducing problems of code integration in a collaborative environment (*Vasilescu et al., 2014*), hence guaranteeing the stability of the code in the mainline. Consequently, CI has found widespread practitioner adoption (*Hilton et al., 2016*), making it a relevant subject of academic study.

Tightly linked to CI (and to the GitHub open source development platform— https://github.com) is the idea of pull request based development (see also Fig. 1 for a schematic overview). In this model, the main repository is not shared with external developers. Instead, prospective contributors fork the central repository and clone it to a local repository. The contributor makes changes to the local repository, and commits their changes there. These local changes are then submitted to the main repository by opening a PR in the central repository. A CI system, such as Travis-CI (https://travis-ci.com), then automatically merges the PR into a test branch and runs the tests to check if the PR breaks the build. Finally, one or more rounds of code review (*Bacchelli & Bird, 2013*; *McIntosh et al., 2014*) are conducted and the integrator decides whether to approve the PR, after which it is merged and closed.

## Does using ci lead to faster pull request delivery?

Note that a CI system is not strictly required for the pull request based development model to be followed. *Bernardo, da Costa & Kulesza (2018)* have studied whether using a CI system, which, as described, automates much of the testing that integrators otherwise would have to do manually, leads to shorter PR delivery times. They collected 162,653 PRs and 7,440 releases of 87 OSS projects using the GitHub API, and addressed the following three research questions:

RQ1: Are merged pull requests released more quickly using CI?

RQ2: Does the increased development activity after adopting CI increase the delivery time of PRs?

RQ3: What factors impact the delivery time after adopting CI?

By applying non-parametric tests to the merge and delivery time of PRs, the authors drew the conclusion for RQ1 that only half of the projects deliver PRs faster after adopting CI, but 71.3% of the studied projects merge PRs faster before using CI. In RQ2, they found that there is a considerable increase in the PR submission, merge and delivery rate, concluding that this may be the reason why projects do not deliver merged PRs faster after adopting CI. They also found that the number of releases per year does not change significantly after CI adoption. In RQ3, they built linear regression models for each project and used the Wald $X^2$ maximum likelihood test to evaluate the explanatory power of a number of different factors. They found that the two variables with the highest explanatory

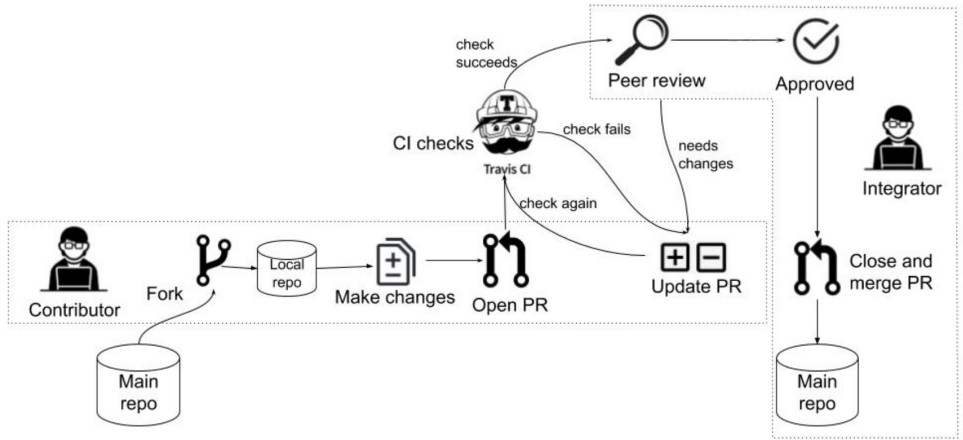

**Figure 1  An overview of the pull request based development model.**

power were both related to the volume of PRs that have to be merged, namely the merge workload (how many PRs are waiting to be merged?) and queue rank (is the PR at the beginning or the end of the merge queue?).

# RELATED WORK

We now discuss previous work in related fields and how the research questions in the study fill in gaps presented in the field.

## CI adoption and pull requests

Previous researchers have investigated the impact of adopting CI in projects in multiple aspects. Most papers agreed that the introduction of CI is beneficial to projects. *Manglaviti et al. (2017)* examined the human resources that are associated with developing and maintaining CI systems. They analyzed 1,279 GitHub repositories that adopt Travis-CI using quantitative methods. The authors found that for projects with growing contributor bases, adopting CI becomes increasingly beneficial and sustainable as the projects age. Further, there is a strong expectation that CI should improve the productivity of projects. *Miller (2008)* analyzed the impact of CI by summarizing their experience with CI in a distributed team environment at Microsoft in 2007. They collected various CI related data in their daily work. Teams moving to a CI driven process can expect to achieve at least a 40% reduction in check-in overhead when compared to a check-in process that maintains the same level of the code base and product quality. *Ståhl & Bosch (2014)* argued based on survey results that build and test automation saves programmer's time for more creative work, and should thus increase productivity. *Stolberg (2009)* argued that CI practices speed up the delivery of software by decreasing integration times. However, not all previous study agree that adopting CI improves productivity. For instance, *Parsons, Ryu & Lal (2007)* found no clear benefits of CI on either productivity or quality.

Related research has shown that the PR based development model is popular in OSS projects. For instance, *Vasilescu et al. (2014)* collected 223 GitHub projects and found

that for 39 of 45 project (87%), builds corresponding to PRs are much more likely to succeed than builds corresponding to direct pushes. *Gousios, Pinzger & van Deursen (2014)* found that 14% of repositories are using PRs on GitHub. They selected 291 projects from the GHTorrent corpus, and concluded that the PR model offers fast turnaround, increased opportunities for community engagement, and decreased time to incorporate contributions.

### CI impact on pull request success and release frequency

Our study focuses on whether CI has an impact on PR delivery time. *Bernardo, da Costa & Kulesza (2018)* have conducted an extensive mining study on this subject (as discussed in more detail in 'Does Using CI Lead to Faster Pull Request Delivery?'). Our present work is a conceptual replication of their paper. *Hilton et al. (2016)* have previously analyzed 34,544 open source projects from GitHub and surveyed 442 developers. The authors found that CI helps projects release twice as often and that when using CI, PRs are accepted 1.6 hours sooner in median. *Vasilescu et al. (2014)* studied the usage of Travis-CI in a sample of 223 GitHub projects written in Ruby, Python, and Java. They found that the majority of projects (92.3%) are configured to use Travis-CI, but less than half actually use it. In follow-up research, they investigated the productivity and quality of 246 GitHub projects that use CI (*Vasilescu et al., 2015*). They found that projects that use CI successfully process, accept, and merge more PRs. This increased productivity does not appear to be gained at the expense of quality. Finally, *Yu et al. (2015)* collected 103,284 PRs from 40 different GitHub projects. They investigated which factors affect PR evaluation latency in GitHub by applying a linear regression model and quantitative analysis. They found that the size of PR and the availability of the CI pipeline are strong predictors or PR delivery time. In later work, the same authors used a linear regression model to analyze which factors affect the process of the pull request based development model in the context of CI (*Yu et al., 2016*). They found that the likelihood of rejection of a PR increases by 89.6% when the PR breaks the build. The results also show that the more succinct a PR is, the greater the probability that such a PR is reviewed and merged quickly.

### Replication studies

The need for conducting more replications of published research is by now rather widely accepted in the software engineering community, as documented through efforts such as the ROSE Festival (held, for instance, at ICSE (https://2019.icse-conferences.org/track/icse-2019-ROSE-Festival) and FSE (https://github.com/researchart/rose3-fse19) in 2019). In general, replication is necessary to increase the trust in any individual piece of research –the results of any one study alone cannot be extrapolated to all environments, as there are typically many uncontrollable sources of variation between different environments (*Shull et al., 2002*). Successful replication increases the validity and reliability of the outcomes observed in an experiment (*Juristo & Gmez, 2012*). *Shull et al. (2008)* distinguish two types of replication studies. In exact replications, the original experimental design is followed as exactly as possible, while a conceptual replication attempts to answer the same research questions using an adapted methodology. We argue that conceptual replications are even

more important than exact ones, as they allow us to control for deficiencies in research design, whereas exact replications mostly validate experiment execution.

However, not all researchers share this excitement about replication studies. *Shepperd (2018)* argued that, due to wide prediction intervals, most replications end up successful anyway. Further, according to *Basili, Shull & Lanubile (1999)*, replication studies in software engineering are particularly difficult to conduct, as experiments in this field usually involve a large number of context variables. Consequently, a systematic mapping study of replications in the software engineering field (*Shull et al., 2008*) concluded that the absolute number of replications is still small, in particular considering the breadth of topics in software engineering. Their study retrieved more than 16,000 articles, from which they selected 96 articles reporting only 133 replications.

Our work is a contribution towards increasing the trustworthiness of research on the impact of CI on PR delivery times. Our replication design combines exact with conceptual replication—we decide to not deviate far from the original design of *Bernardo, da Costa & Kulesza (2018)*, and also largely follow their style of presentation, while at the same time addressing the methodological concerns we had with their original work.

## METHOD

The goal of the present study is to replicate and extend the results from earlier work presented in 'Does Using CI Lead to Faster Pull Request Delivery?'. We now discuss our scientific methodology and the data that has been used. Fig. 2 provides a schematic overview. For RQ1, RQ2.1, and RQ2.2, the data set from the original study is re-used. For RQ3.1 and RQ3.2, two new data sets are collected from GitHub. For RQ1, the original statistical methods are re-used. For RQ2.1, an alternative analysis approach (RDD) is employed. For RQ2.2, the same method is extended with an additional analysis variable (the point in time in the release cycle when a PR is submitted, "come-in time"). For RQ3.1, all analyses are applied to the new data sets. For RQ3.2, we only apply non-parametric tests, as our findings do not warrant applying the rest of the analyses to this data set. All data as well as the necessary analysis scripts are publicly available on GitHub https://github.com/radialine/Do-Open-Source-Projects-Deliver-Pull-Requests-Faster-Using-CI.

### Study subjects and data collection

As depicted in Fig. 2, our study relies on three different sets of study objects –the *original data* provided by the authors, a set of new projects collected using the same methodology (*new data*), and a control group consisting of projects collected using the same methodology, but which, crucially, have to the best of our knowledge never adopted CI (*control data*). Basic information about the three data sets is contained in Table 1. The collection procedure is further described below.

#### *Original data*

We re-use the data that *Bernardo, da Costa & Kulesza (2018)* have made available online https://prdeliverydelay.github.io/#datasets. However, for a subset of our analysis, we need

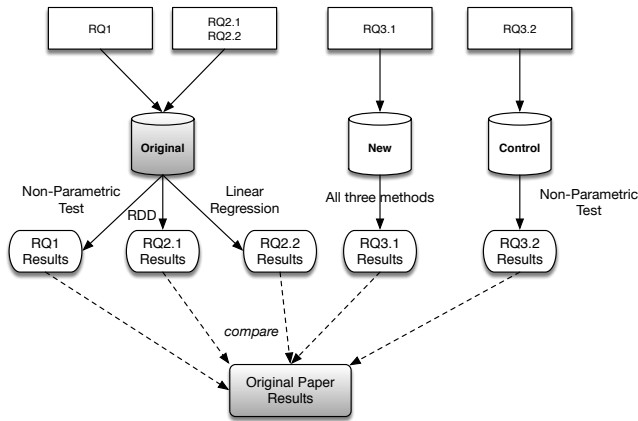

**Figure 2   Overview of study methodology and used data.** Shaded elements are re-used from *Bernardo, da Costa & Kulesza (2018)*.

additional information not contained in the original data (e.g., the exact point in time when a PR was merged). This information was collected directly through the GitHub API.

### New data

For collecting a new data set, we largely follow the process originally used by *Bernardo, da Costa & Kulesza (2018)*, which in turn was inspired by *Vasilescu et al. (2015)*. We identify the 800 most highly-starred projects on GitHub written in Java, Python, PHP, Ruby, and JavaScript. This leads to a total of 2763 unique projects (projects that use multiple of these languages are counted only once). We discard all projects that are not using Travis-CI, as well as all projects that were already contained in the original data set. We further exclude all projects that have less than 100 merged PRs before or after CI adoption. That is, we only consider projects that have had reasonable development activity before and after adopting CI. Finally, we also discard toy projects, tutorials, and other similar projects that are not intended to be deployed to production. This leaves us with 54 projects, for which we then collect PR and release data using git and the GitHub API.

### Control data

We use the same process as for *new data* to collect a control group, with the key difference that we discard all projects for which we can tell that they are, or have been, using any CI system, leading to 28 projects. Note that this data set is smaller, as, given the prevalence of CI, it is difficult to find high-profile projects with similar characteristics to the projects in the other two data sets which never adopted CI in their lifetime.

## Analysis methods

As shown in Fig. 2, we use three different statistical methods in our study. We replicate two of the methods used in the original study, and introduce a third, new, method (regression discontinuity design, RDD).

| Table 1 Basic data set statistics. | | |
|---|---|---|
| **Data set** | **# of projects** | **Total # of PRs** |
| Original data | 87 | 162.653 |
| New data | 54 | 84.487 |
| Control data | 28 | 47.519 |

### *Methods re-used from the original study*

In line with the original work, we use non-parametric hypothesis testing (Mann-Whitney-Wilcoxon, MWW) for testing whether there is a statistically significant difference in pull request delivery time before and after CI introduction. MWW is used as data normality could not be assumed. MWW is used in conjunction with Cliffs delta to measure effect sizes, using the standard threshold values as defined by *Romano et al. (2006)*. Additionally, we use a multiple regression model fitted with ordinary least squares to identify which factors best explain a dependent variable (delivery delay, in our case). We use the Wald $X^2$ maximum likelihood test to evaluate the explanatory power of each independent variable.

### *RDD*

Due to our concern that the original study did not properly control for changes in PR submissions and PR delivery time that are independent of CI (due to, for instance, project growth or other project lifetime related factors), we extend the original work with an additional statistical method, RDD, as inspired by the work of *Zhao et al. (2017)*. RDD is a fairly old idea firstly proposed by *Thistlethwaite & Campbell (1960)*, which is seeing a renaissance in recent years (*Imbens & Lemieux, 2008*). It is a quasi-experimental pretest-posttest design that elicits the causal effects of interventions by assigning a cutoff above or below when an intervention is applied (CI introduction, in our case). The assumption of RDD is that the trend continues without changes if the intervention does not exist. We would conclude that CI had a significant impact if there is an obvious discontinuity around the cutoff point (the point in time when the intervention has been applied).

A core question when applying RDD is which model(s) to use for fitting the data before and after the intervention. In this study, four models of RDD are used, as sketched in Fig. 3. The linear model with common slope assumes that the data before and after the intervention can be fit using the same linear regression model (shifted by a constant), while the linear model with different slopes only assumes that both sides can be fit by any linear regression. The non-linear model assumes that at least one side requires fitting using a non-linear regression. Finally, local linear regression performs exactly that using the Imbens-Kalyanaraman optimal bandwidth calculation.

## RESULTS

We now discuss the results for each research question. Given that this is a replication study, a particular emphasis will be put on comparing our results to *Bernardo, da Costa & Kulesza (2018)* and evaluating to what extent the results therein still hold.

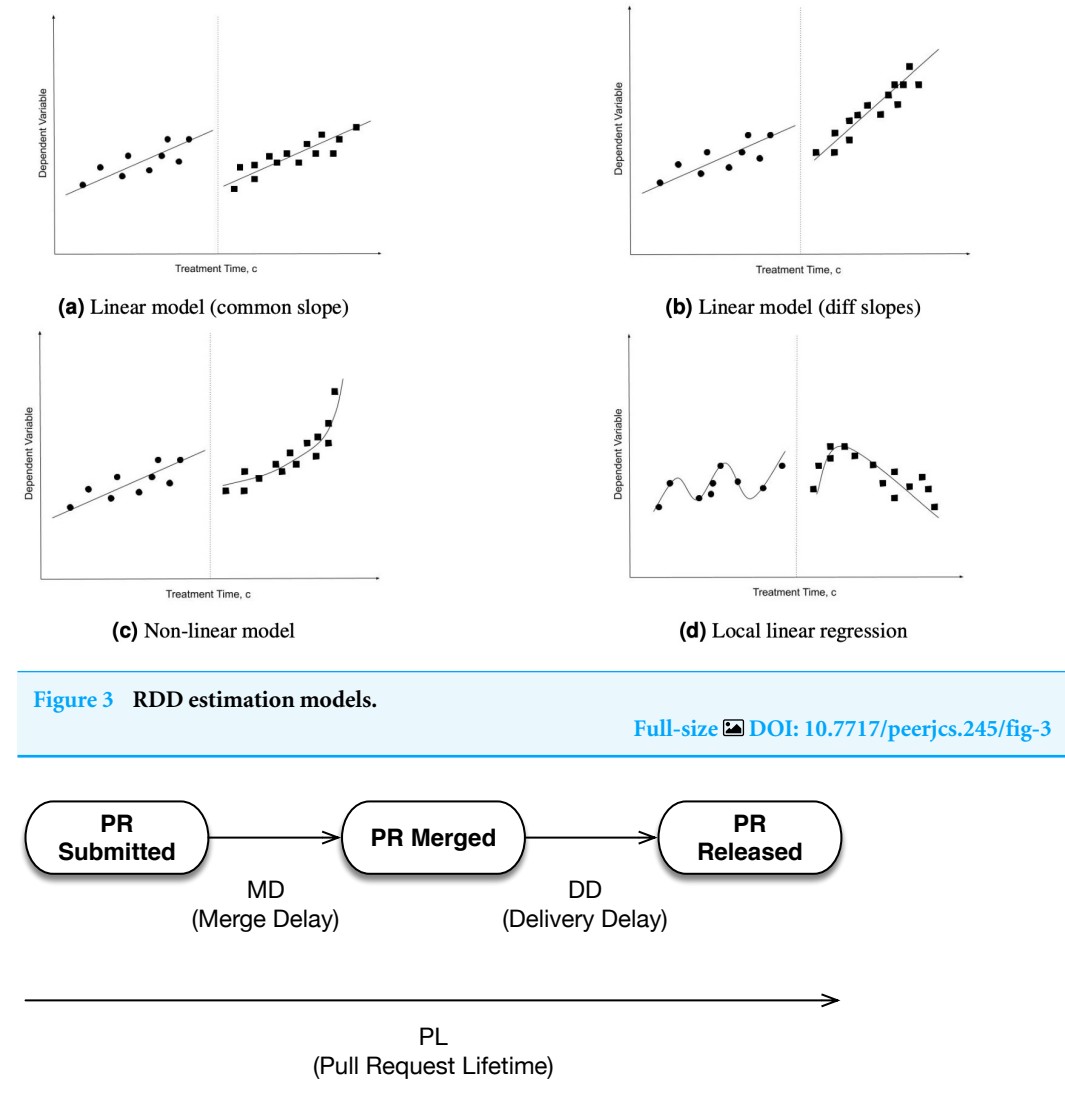

**Figure 3** RDD estimation models.

**Figure 4** **Graphical overview of the evaluated metrics DD, MD, and PL.** Adapted from *Bernardo, da Costa & Kulesza (2018)*.

## RQ1 –exact replication

As a first step in the study, we conducted an exact replication of *Bernardo, da Costa & Kulesza (2018)*, based on the data that the authors provide. This was deemed necessary as a first step of validation, but also to acquire the necessary in-depth knowledge about the original study's design choices.

RQ1 of the original study investigated the impact of adopting CI on the delivery time of PRs. They analyzed three metrics, which are *delivery delay* (DD, days between when a PR got merged and when it was released), *merge delay* (MD, days between when a PR was submitted and when it was merged), and *pull request lifetime* (PL). A visual overview of these metrics and what they mean in the PR lifecycle is presented in Fig. 4.

**Table 2  Results of the exact replication of RQ1 in *Bernardo, da Costa & Kulesza (2018)*.**

|     |             | Faster with CI [% of Projects] | Stat. Different [% of Projects] |
|-----|-------------|--------------------------------|---------------------------------|
| DD  | Original    | 51.4%                          | 82.8%                           |
|     | Replication | 47.9%                          | 83.9%                           |
|     | Difference  | **3.5**                        | **1.1**                         |
| MD  | Original    | 27%                            | 72.4%                           |
|     | Replication | 29.4%                          | 78.2%                           |
|     | Difference  | **2.4**                        | **5.8**                         |
| PL  | Original    | 48.4%                          | 71.3%                           |
|     | Replication | 52.4%                          | 72.4%                           |
|     | Difference  | **4**                          | **1.1**                         |

After carefully studying the original paper and limited follow-up discussion with the authors through private communications, we are able to reproduce their results. Table 2 contrasts the original results with the results of our exact replication. We report on the percentage of projects for which each of these metrics improved after introducing CI (i.e., handling PRs became faster) and the percentage of projects for which there is a statistically significant difference (in any direction). Cliff's delta effect sizes for the latter metric vary between 0.2 and 0.3 (i.e., a small effect size), except for the changes in pull request lifetime, where we observe medium or even large effect sizes for a majority of projects.

It is interesting to note that even though we used the same methods on the same data, we were not able to achieve entirely identical results (differences between 1.1 and 5.8 percentage points). We speculate that the observed differences may be due to undocumented data cleaning procedures or updates to the publicly available data set. However, given that the main findings of the study remain unchanged, we nonetheless consider the replication successful.

RQ2 in the original study then tried to find the reason for this phenomenon. The authors compare the number of submitted, merged, and released PRs before and after CI adoption. We again replicate this analysis, leading to the results depicted in Fig. 5. For this analysis step, our results are virtually identical to what has been presented in *Bernardo, da Costa & Kulesza (2018)*. We observe that after CI was adopted, the number of submitted, merged and released PRs per release increases statistically significantly with medium effect sizes. Interestingly, the release frequency does not change statistically significantly after adopting CI.

---

**Box 1.   Summary and Lessons Learned.**

We were able to conduct an exact replication of the original paper, with minor differences in the results (between 1.1 and 5.5 percentage points). All main results of the original study are confirmed. This analysis indeed supports that only about half the projects deliver PRs faster (with a small effect size) after introducing CI, but less than a third of projects improves how fast they merge PRs (again with a small effect size). While projects do not seem to release more frequently, they can handle more PRs per release after CI adoption.

---

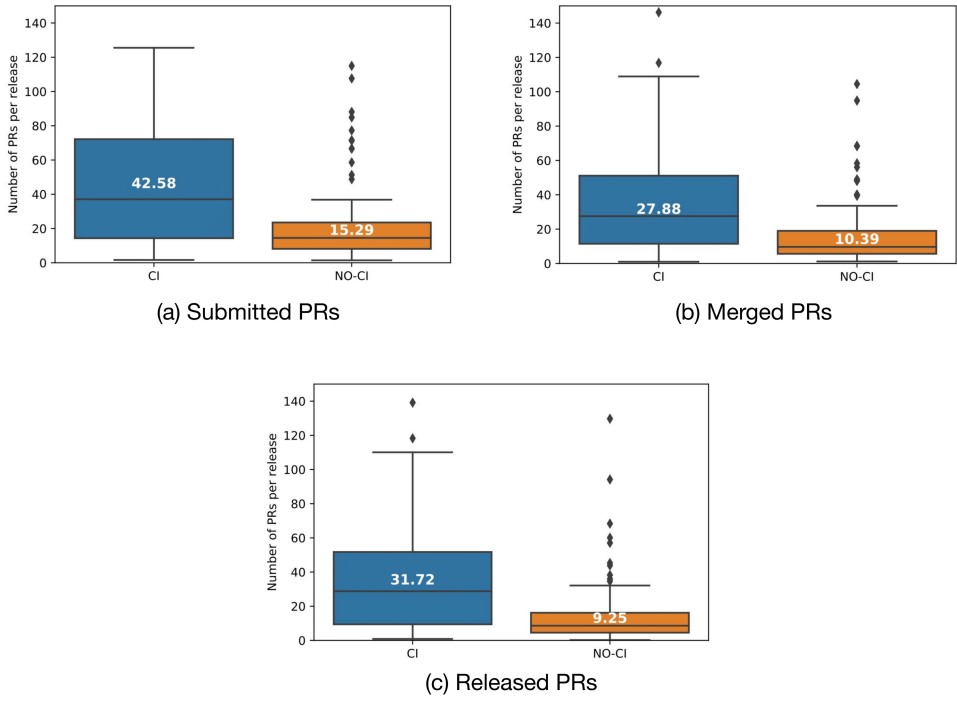

(a) Submitted PRs

(b) Merged PRs

(c) Released PRs

**Figure 5** **Comparison of merged and released PRs per release before ("NO-CI") and after ("CI") introduction of CI.**

## RQ2 –conceptual replication

We now discuss the two additional analysis steps we have introduced in our study in comparison to the original work.

### Application of RDD (RQ2.1)

In the first step of our conceptual replication, we use RDD to analyze whether there are gradual changes in PR delivery time over the lifetime of projects, independently of CI introduction. However, an initial visual inspection of both, DD and PL, reveals that these metrics follow a clear pattern that is independent of CI introduction. Figures 6 and 7 depict this for two example projects (`mantl/mantl` and `mozilla-b2g/gaia`).

Virtually all 87 projects in the original data set follow a similar pattern, indicating that these metrics are to a large degree dominated by when in the release cycle a PR comes in –PRs merged shortly after a release need to wait for the next release to roll around, while PRs merged shortly before a release get released much quicker. It is unlikely that the introduction of CI has much direct impact on this. It should be noted that this is true even for PL, which represents the entire delivery time of a PR (i.e., the time it takes maintainers to merge a PR plus the time the PR then waits to get released). Hence, it seems unlikely that the introduction of CI can impact this end-to-end delivery time of a PR by much. This also explains why we, similar to the original study, observe primarily differences with small effect sizes in RQ1. Ultimately, the end-to-end delivery time is presumably much

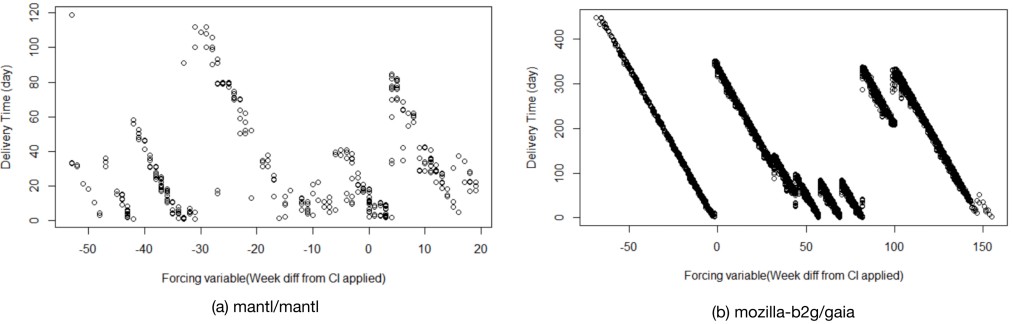

**Figure 6** **Visual inspection of metric *delivery delay* (DD) for two example projects.** (A) mantl/mantl, (B) mozilla-b2g/gaia. The x-axis represents project lifetime in weeks, with point 0 being the RDD cutoff point (i.e., the time when CI has been adopted in the project).

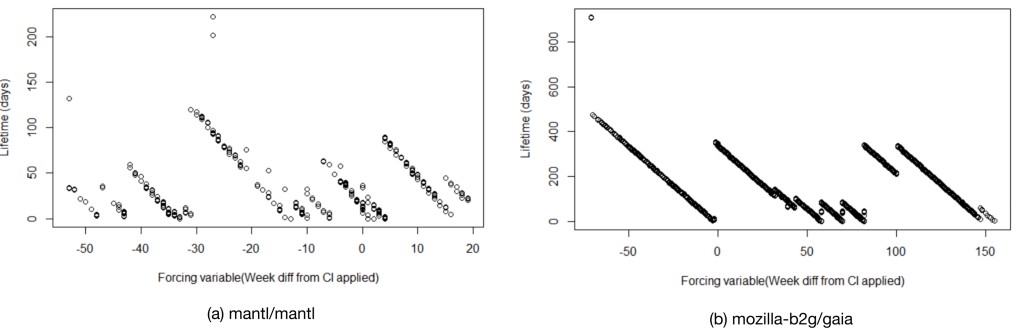

**Figure 7** **Visual inspection of metric *pull request lifetime* (PL) for two example projects.** (A) mantl/-mantl, (B) mozilla-b2g/gaia. The x-axis represents project lifetime in weeks, with point 0 being the RDD cutoff point (i.e., the time when CI has been adopted in the project).

more dependent on how frequently a project releases than on whether a CI system is used, which we have established in RQ1 to not be impacted by CI adoption.

However, no such pattern exists for the third metric, MD. Hence, we attempt to apply all four RDD models described in 'Analysis Methods'. The data of each project is divided into two buckets separated by the cutoff point (when CI was adopted), and one model for each bucket is fit. Fig. 8 shows the fitted models of project `boto/boto`. In the first three models, the red and blue lines fit data after and before the intervention respectively.

It is evident that neither the two linear models (Figs. 8A and 8B) provide sufficient fit to accurately represent the data for `boto/boto`. Indeed, the linear or non-linear models never achieve an $R^2$ value higher than 0.35 for any of the 87 projects. The local linear regression model depicted in Fig. 8D provides a better, albeit still very noisy, fit to the data. Hence, we conclude that there is no, or at least no particularly relevant, "natural trend" of MD getting faster or slower over time in any of the projects. Hence, we consider our original concern with the work of *Bernardo, da Costa & Kulesza (2018)* (that projects may just naturally get faster or slower over time) to be unsupported.

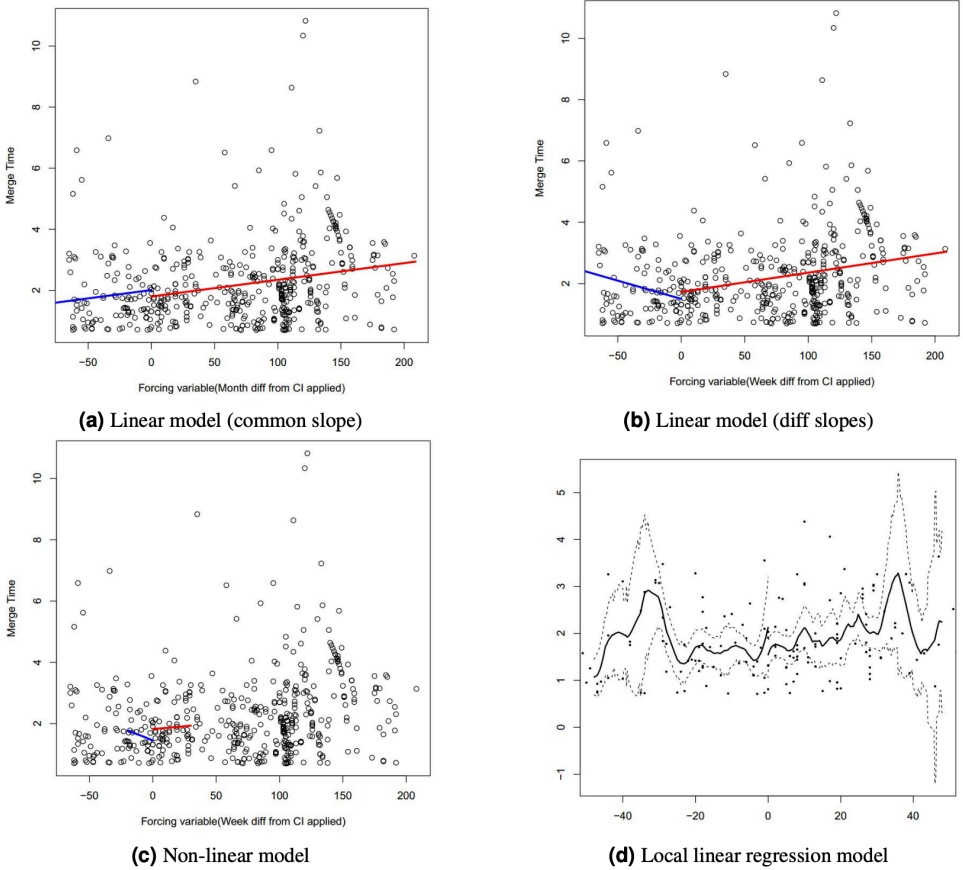

**Figure 8** **Four RDD models fit for project boto/boto.** (A) Linear model (common slope). (B) Linear model (diff slopes). (C) Non-linear model. (D) Local linear regression model. The x-axis represents project lifetime in weeks, with point 0 being the RDD cutoff point (i.e., the time when CI has been adopted in the project).

### Evaluation of "Come-In Time" as Predictor of PR Delivery Time (RQ2.2)

In an attempt to explain what exactly impacts the end-to-end lifetime of a PR (PL), the original study built a multiple regression model based on 13 different variables (related to characteristics of the project, the PR submitter, and the PR itself). They found that three metrics (merge workload, queue rank, and, to a lesser degree, the contributor) had significant explanatory power with regards to PL. Before CI adoption, the merge workload has the highest explanatory power, which changes to the queue rank after adoption. Based on our previous findings, we speculate that in fact the most important predictor of end-to-end PR delivery time may be when in the release cycle a PR has been merged. We refer to this new factor as "come-in time", and provide a schematic overview of its definition in Fig. 9.

We re-use the original methodological setup (regression analysis using ordinary least squares), but use the variables sketched in Table 3. We remove all variables which had an explanatory power close to 0 in the original study, leaving us with 6 potential factors

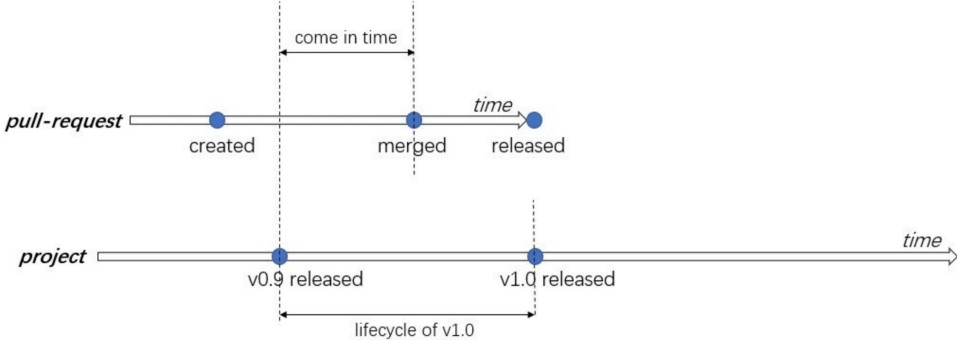

**Figure 9   Definition of the factor "come-in time".**

**Table 3   Description of all variables used in the regression model.** The first 6 variables are re-used from *Bernardo, da Costa & Kulesza (2018)*, the last variable has been newly introduced in our study.

| Variable | Definition |
|---|---|
| *Variables From Original Study* | |
| Number of Activities | An activity is an action to a PR conducted by a GitHub user, e.g., labeled, assigned, etc. It is assumed that a large number of activities may lead to longer delivery times. |
| Merge Time | The time between when a PR was created and when it was merged by an integrator (MD). |
| Contributor Experience | The number of released PRs that were created by the same author. We speculate that contributions by an experienced contributor may be evaluated less critically, and hence may be delivered faster. |
| Contributor Integration | The mean delivery time in days of the past PRs submitted by this contributor. If past PRs were released quickly, then the next PR submitted by the same person may also be released rapidly. |
| Merge Workload | The number of PRs waiting to be merged at the time when the PR was submitted. We speculate that, as the time and energy of integrators is limited, the workload of an integrator may have an impact on delivery times. |
| Queue Rank | This variable represents the order of the merged PRs in a release cycle. A merged PR might be released faster or slower depending of its position in the merge queue. |
| *New Variable* | |
| Come-in Time | The time in days between the time when a PR got merged and the time of the last release (see also Figure ??). This new variable is motivated by our previous findings. |

of influence ("merge workload", "queue rank", "contributor experience", "contributor integration", "number of activities", and "merge time"). We add the new variable "come-in time" to this set.

From these variables, we build two regression models for each project (before and after CI adoption), and evaluate the $R^2$ metric for each model. $R^2$ represents how much of the variability in the data can be explained using the model. Following *Bernardo, da Costa*

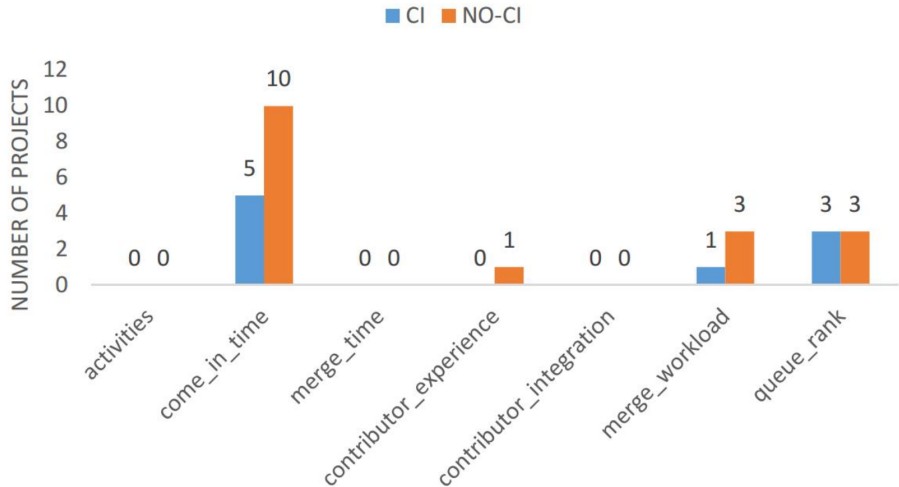

**Figure 10** **Bar chart plotting the variables with the highest explanatory power for each project, before ("NO-CI") and after ("CI") CI adoption.** Only projects for which a regression model with $R^2 > 0.5$ could be trained are considered.

*& Kulesza (2018)*, we only accept models with $R^2 > 0.5$ as sufficiently accurate. Prior to CI adoption, the models for 17 of 87 projects (19.8%) have $R^2$ values higher than 0.5 (median of these is 0.58). After CI adoption, we achieve only 9 valid models (10.5%), with a median $R^2$ value of 0.57. This is in line with our previous findings, and indicates that PR delivery time is in general rather unpredictable, and unlikely to depend on any single factor.

Figure 10 depicts for how many projects (among those for which a model with $R^2 > 0.5$ could be found) each variable is the one with the highest explanatory power, as measured through the Wald $X^2$ maximum likelihood test. Our newly proposed variable "come-in time" indeed outperforms all variables from the original study. This further supports that the factor most important to the end-to-end delivery time of a PR is whether it has been merged close in time to the next release. It is also noticeable that all variables related to the nature of the PR or the contributor are less relevant than process- and project-oriented metrics, such as when a PR comes in, which position in the merge queue it has, or how large the merge workload currently is.

It needs to be noted that there is a high correlation between the new metric "come-in time" and "queue rank", one of the metrics in the original study, in a subset of the projects. Namely, in 19 of 87 projects (22%) the correlation between these metrics is larger than 0.7 prior to introducing CI, and in 21 of 87 projects (24%) after CI introduction. For the remaining projects, there is a correlation between these metrics, but it is less pronounced.

> **Box 2.** Summary and Lessons Learned.
>
> Applying RDD to the original data set primarily revealed that two of the three analyzed metrics (DD and PL) follow very clear patterns, namely that they depend to a large degree on the time until the next release. Consequently, when in the release cycle a PR is merged is the best predictor of delivery time (PL). The merge delay MD does not follow such a pattern. We did not observe in any project that MD would trend up- or downwards independently of CI adoption, alleviating our original concern with the original study. However, our experiments also confirm the result from the original study that the delivery time is generally difficult to predict, as indicated by the low $R^2$ metric of the regression models of most projects.

### RQ3 –Generalizability

So far, we have applied all analyses to the data set also used in the original study. Now we turn towards evaluating whether the previous findings are specific to the used data.

#### Analysing new data (RQ3.1)

In a first step, we evaluate the generalizability of our findings by collecting 54 new projects (which have also adopted CI), and conducting the same analyses as presented in 'RQ1—Exact Replication' and 'RQ2—Conceptual Replication'.

We firstly again evaluate how many projects improved DD, MD, and PL, and used a MWW test to evaluate statistical significance. The results of this analysis are provided in Table 4, which also provides our own results from RQ1 as a point of comparison. We observe that the results are not fundamentally different, although we observe 10 to 14 percent points difference in selected results (particularly related to the delivery delay DD). Effect sizes are small, as also observed for the original data. A replication of our analysis of submitted, accepted, and released PRs confirms our findings that projects statistically significantly increase their development activities after adopting CI (with medium effect size), but we can again not find a statistically significant change in the number of releases. Finally, the re-execution of RDD (RQ2.1) on the new data yields similarly comparable results. A deeper discussion of this aspect is omitted here for reasons of brevity, but can be found in *Guo (2019)*.

An interesting result is found when fitting regression models, as discussed for RQ2.2, to the new data. For 54 projects, only 2 models (3.7%) trained on data after CI adoption and 5 models (9.3%) for data before CI adoption achieve an $R^2$ metric higher than 0.5. It remains unclear why the regression approach works even less well on the new than on the original data. However, given that $R^2$ values were generally low even for the original data, this result may ultimately just stress that predicting delivery times is difficult at the best of times.

#### Analysing a control group (RQ3.2)

So far, we have experimented only with projects that actually adopted CI at some point in the project's lifetime. We now turn towards analysing our control group of comparable

**Table 4  Results of a re-analysis using a new data set.**

| | | Faster with CI [% of Projects] | Stat. Different [% of Projects] |
|---|---|---|---|
| DD | Original data | 47.9% | 83.9% |
| | New data | 58% | 98% |
| | Difference | **10.1** | **14.1** |
| MD | Original data | 29.4% | 78.2% |
| | New data | 19.5% | 80.4% |
| | Difference | **9.9** | **2.2** |
| PL | Original data | 52.4% | 72.4% |
| | New data | 46.5% | 79.6% |
| | Difference | **5.9** | **7.2** |

**Table 5  Results of a re-analysis using a control group of projects which never introduced CI.**

| | | Faster with CI [% of Projects] | Stat. Different [% of Projects] |
|---|---|---|---|
| DD | Original data | 47.9% | 83.9% |
| | Control data | 59.2% | 96.4% |
| | Difference | **11.3** | **12.5** |
| MD | Original data | 29.4% | 78.2% |
| | Control data | 28% | 89.3% |
| | Difference | **1.4** | **11.1** |
| PL | Original data | 52.4% | 72.4% |
| | New data | 40% | 89.3% |
| | Difference | **12.4** | **16.9** |

projects which, as far as we can observe, have never adopted CI. One challenge in this context is what point in the project's history to use as cutoff for analysis. From analysing the 87 projects in the original data set, we learn that these projects, on average, introduce CI after 38.2% of the lifetime of the project in days (median 38%, variance 8.5). Hence, we decide to introduce a "mock-ci-timepoint" for the projects in the control group that corresponds to 38% of their lifetime. Intuitively, this is the point in time when these projects would have, on average, adopted CI (if they ever did).

A comparison of the results achieved for this control group with the results achieved for the original data set is provided in Table 5. Note that in this case "Faster with CI" for the control group should be interpreted as "faster after the mock-ci-timepoint" of 38%.

The results of this analysis indicate that we do observe (slightly) larger differences between the original test group and the control group than what we have observed for the two different test groups in RQ3.1 (cp. Table 4). This supports the conclusion that the introduction of CI has some modest impact on these numbers.

However, when analyzing the number of submitted, merged, and released PRs, we observe that there is no difference between before and after the (mocked) CI introduction. This is visualized in Fig. 11. Statistical testing does not reveal any differences before and after the mocked CI introduction for any metric.

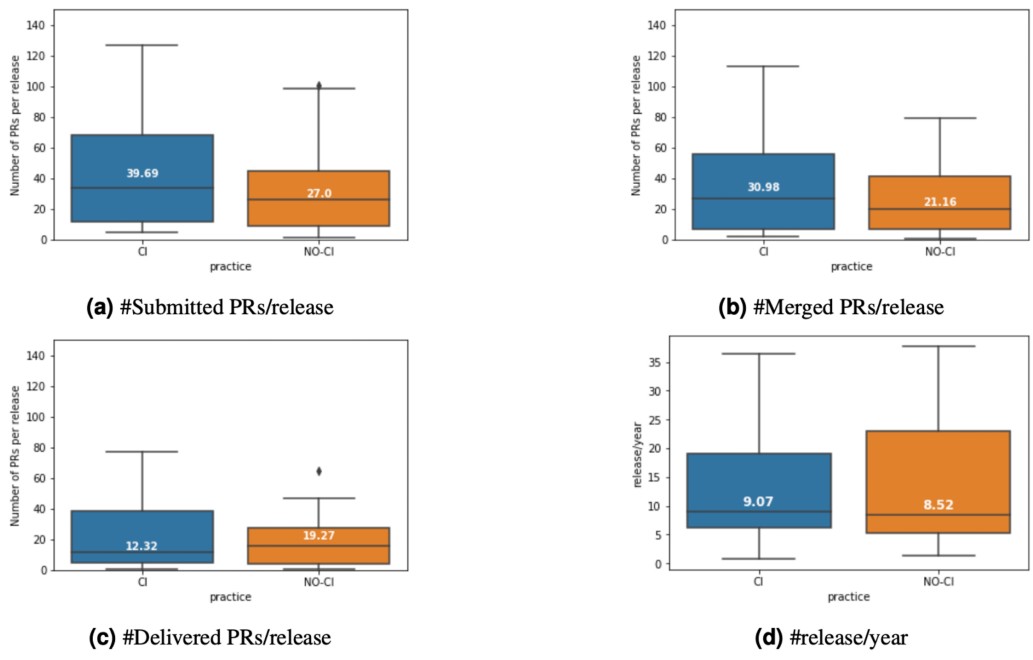

**Figure 11** Comparison of merged and released PRs per release before and after CI inroduction for a control group of projects that never introduced CI. (A) #Submitted PRs/release; (B) #Merged PRs/release; (C) #Delivered PRs/release; (D) #release/year.

Hence, we support the argument by *Bernardo, da Costa & Kulesza (2018)* that the introduction of CI seems to have a (minor) impact on PR delivery time of projects. However, projects in both test groups manage to handle considerably more PRs per release after CI adoption, while we have not observed any statistically significant increase in the control group. Hence, we conclude that projects do not so much speed up handling individual PRs, but rather manage to handle considerably more PRs per release after adopting CI.

---

**Box 3.** Summary and Lessons Learned.

Applying our analyses to new data sets allowed us to evaluate to what extent the effects observed so far are due to specifics of the data collected by *Bernardo, da Costa & Kulesza (2018)*. When analyzing a new data set collected using the same methodology, we have observed results that are in the broad strokes similar to the original findings, although we have observed differences up to 14 percentage points for individual metrics. When analyzing a control group of projects that never adopted CI, we found results not unlike to the results of the new test group, indicating that the small size effects we observed in RQ1 may be independent of CI introduction. However, we have observed that both test groups handle more PRs after CI adoption with medium effect size, while we have not observed a statistically significant increase for the control group. This leads us to believe that projects may not actually handle individual PRs

---

> (much) faster after CI adoption, but they are able to handle considerably more PRs per release.

## THREATS TO VALIDITY

This section addresses potential threats to the validity of our replication and overall results.

**Construct validity** Construct validity describes threats related to the design of the study. To a large degree, we have chosen the same data collection procedures, statistical methods, and analysis techniques that were already present in the original study. This was done by design, so as to keep our replication easily comparable to the original study. However, this also means that any limitations inherent in the original study design are still present in our replication (with the exception of those that we explicitly chose to address as part of our conceptual replication). For the construction of our control group, there are two related threats. (1) Even though we carefully attempted to determine whether a candidate project for the control group does indeed not use any CI system, it is not always feasible to determine this from an outsider's point of view (e.g., a company-backed OSS project may use a CI system within the company, which is not mentioned on the GitHub page). (2) Even though we attempted to keep the control group as similar in characteristics to the original study objects as possible, the mere fact that these projects have chosen to not adopt CI may already hint at deeper differences in mindsets, processes, and project goals than what is visible from GitHub metrics alone. These differences may also account for some of the different results we have observed. Further, our control group is considerably smaller than the original data set (28 versus 87 projects).

**External validity** External validity concerns to what extent the findings of the study still hold under in more generalized circumstances. Part of our replication was specifically to investigate a data set of 52 new projects which adopted CI, and 28 projects which are not using CI. However, we used the same data collection procedure and sampling methods to select these projects. Hence, our replication does not aim to, and cannot, answer the question if the observed results are specific to OSS software, to high-profile projects, or to projects written in the Java, Python, PHP, Ruby, or JavaScript programming languages. Further, it should be noted that we only consider projects that make use of Travis-CI. Hence, it remains an open question to what extent our results also generalize to projects using other CI systems, such as GitLab https://about.gitlab.com or Jenkins https://jenkins.io.

**Internal validity** Internal validity questions to what extent the study is able to draw correct conclusions, and does not fall prey to, for instance, confounding factors. One of the key motivations of our replication was to evaluate whether normal changes in projects over the lifetime of the project may be responsible for the effects observed in the original study. This concern was alleviated in our replication. However, other confounding factors may still remain relevant. Particularly concerning in this regard is that our evaluation of a control group of projects that never applied CI has shown results that, ultimately, were not fundamentally different than what we observed for a new data set of CI-using projects.

Hence, we see the need for more work to fully establish the effects of adopting CI in OSS projects.

## CONCLUSIONS

In this work, we replicated an original study by *Bernardo, da Costa & Kulesza (2018)* that attempted to answer the question whether OSS projects deliver PRs faster after adopting CI. Our replication was motivated by limitations in the original study design, which did not account for changes in PR delivery time independent of CI introduction. We conducted an exact replication of the original work, analyzed the original data using a different statistical procedure (RDD), and extended the original multiple regression model using a new variable ("come-in time"). Further, we analyze two new data sets, a new set of study subjects that adopted CI and a control group of projects that did not.

We were able to replicate the original findings. Our analysis using RDD has not shown any evidence of growth of PR delivery times independent of CI introduction, and our analysis of control group data has revealed that projects which never adopted CI do not see the same increase in submitted, merged, and released PRs as seen for CI-using projects. However, our study also confirms that the impact of CI on the delivery time for an individual PR is only minor. This is in line with the original study, which has also reported primarily small statistical effect sizes. We further find that, before as well as after CI adoption, the best predictor of PR delivery times is when in the release cycle a PR is merged. This indicates that, ultimately, projects need to increase the number of releases to speed up PR delivery times rather than adopt CI. However, the number of releases appears to be largely independent of whether or not a project adopts CI.

## ACKNOWLEDGEMENTS

This work has been conducted as a master project while the first author was a student at Chalmers University of Technology.

### Funding

This work has received financial support by the Swedish Research Council VR under grant number 2018-04127 (Developer-Targeted Performance Engineering for Immersed Release and Software Engineers). The funders had no role in study design, data collection and analysis, decision to publish, or preparation of the manuscript.

### Grant Disclosures

The following grant information was disclosed by the authors:
Swedish Research Council VR under grant number 2018-04127 (Developer-Targeted Performance Engineering for Immersed Release and Software Engineers).

### Competing Interests

Philipp Leitner is an Academic Editor for PeerJ Computer Science.

## Author Contributions

- Yunfang Guo conceived and designed the experiments, performed the experiments, analyzed the data, prepared figures and/or tables, performed the computation work, authored or reviewed drafts of the paper, approved the final draft.
- Philipp Leitner conceived and designed the experiments, prepared figures and/or tables, authored or reviewed drafts of the paper, approved the final draft.

## Data Availability

All data is available on GitHub: https://github.com/radialine/Do-Open-Source-Projects-Deliver-Pull-Requests-Faster-Using-CI.

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
