# Peer review of "Studying the impact of CI on pull request delivery time in open source projects—a conceptual replication"

_PeerJ Computer Science, doi:10.7717/peerj-cs.245_

## Round 0.1 · original submission · Minor Revisions

This paper seeks to assess the time it takes to merge in a new pull request in an open source project.

The starting point is an existing study (Bernardo et al) which established that the use of continuous integration can speed up merge time (albeit with small effect sizes).

The present paper extends these results in three ways:

1. It conducts an exact replication (reproduction) of the original results
2. It finds that proximity to release deadlines has a (more) important effect on integration times, and (hence) that "come in time" is a good predictor for integration time
3. By carrying out a study in a control group, the paper concludes that projects do not so much speed up handling individual PRs, but they manage to handle considerably more PRs per release after adopting CI.

These are important findings. They also serve to illustrate the importance of open science and open data, taking a replication as a starting point.

We invited two external reviewers, as well as one author of the original paper you replicated -- Daniel Alencar de Costa, in line with the PeerJ policy on replications.

All three are positive about the paper. Reviewer two has various smaller remarks. Reviewer three suggests two discussion items to be added, relating to projects without Continuous Integration (NOCI), and potential correlations between the proposed new metric and the existing metrics.

We ask you to address these remarks, and submit a minor revision of this paper.


Details:

"we consider the replication largely successful" -- can you make explicit in what way?
"to what extend" -> to what extenT

Reviewer 1 ·

Basic reporting

Basic reporting of the submitted article is in general good. The text is clear and concise. One place where the authors can potentially improve the write-up is in the reference section. I am not sure if this formatting is supported by PeerJ but as a reader, it feels odd finding references work.

I see one typing error (see line number 258). Please revisit the document for typing errors.

Experimental design

The objective of the paper is presenting a conceptual replication on the effects of adopting continuous integration on pull request delivery time. This study builds on the original work by Bernardo et al. (2018). This paper replicates the original study and includes a factor that can potentially explain the observation. This paper further tests the observation on a new set of projects to test the usefulness of conclusions.

The study seems to be meticulously designed with potential threats sufficiently addressed in threats to validity. The choice of statistical tests seems appropriate too.

Validity of the findings

The study replicates the original study testing the findings of Bernardo et al (2018). The study furthers adds a factor that can potentially explain pull request delivery times. Using original projects as well as new projects (including controls) the study suggests that the results of the original study holds and more generally predicting PR delivery time is difficult.

As a conceptual replication, study does justice in stepwise replicating the original study but also providing deeper context into the relations.

·

Basic reporting

This paper presents a replication study analyzing the impact of continuous integration (CI) on pull request delivery time.
First, the paper presents an exact replication of previous work, and then it presents a conceptual replication as well as describing the generalizability of the results.
Overall, the paper asks and answers five research questions on this topic.
The paper does a good job of situating this study amid previous work, both in discussing the research under replication, as well as other related work.
The article was well structure, and contains links to the supporting data, which is available on GitHub.

Unfortunately the paper does have a few minor presentation flaws:
Page 1, Line 28 in the abstract: comparabe -> comparable
Page 5, Line 186, "and merge more PR" -> more PR's
Figure 10 is hard to read, and could be much easier to parse if it were expanded to fill the available space.

Experimental design

This work is a well designed replication experiment. By producing an exact replication as well as a conceptual replication, there is much higher confidence in the findings.
The conceptual study also clearly address the concern of the previous work, which had not accounted for the potential that projects might naturally speed up at delivering over time, independent of CI.

One minor improvement would be if the paper were to describe the criteria used to determine if a project were worth being filtered out. Currently the paper states that "we also discarded toy projects, tutorials, and other similar projects not intended to be deployed" It would be helpful if they stated how these projects were determined to be in this category.

Validity of the findings

This paper does a great job of answering the research questions posed in a clear and data-driven way, while also acknowledging the potential threats to validity.
The findings are backed up by the data, and the data is readily available to be examined and used by future researchers.
Additionally, this paper provides a valuable replication of exiting work, and by mostly validating the original findings, they provide much higher confidence.

Additional comments

Overall this paper provides a valuable replication study to evaluate the impact of CI on code delivery.

·

Basic reporting

The paper is very well written and structured. The provided background
and used terminology are sound. Finally, the tables and pictures are very
instructive and help to convey the ideas discussed in the paper.

Experimental design

The paper aims to perform a conceptual replication of the work performed by
Bernardo et al, 2018. The identified gaps are (i) the need for replication
studies in empirical software engineering and (ii) the opportunities for
improving the methodology employed by us (i.e., Bernardo et al. 2018).
The methodology applied in the paper is very well motivated and sound.

Validity of the findings

The overall validity of the paper is sound and contains some opportunities for improvement (as expected in a scientific paper).

Additional comments

I am glad with the opportunity to review this paper as it presents a conceptual
replication of our work (Bernardo et al, 2018). The analyses presented in this
work are not only reassuring for me, but also provide learning opportunities
for future research. I thank the authors for that.

Regarding the design choices of the study, I think the paper can be mainly improved in two aspects:
1) the collection of the NOCI projects and 2) the correlation & redundant analyses before
introducing the 'come-in-time' metric.

- NOCI projects) Sizílio et al (2019) observed that some projects which do not
use Travis-CI do actually use other tools for implementing CI. However, the
CI implementation of these projects is not as easily identified as checking
whether a project has builds on the Travis-CI server. Indeed, Sizílio et al (2019)
sent a little survey for GitHub developers asking whether they have used other
CI implementations (i.e., other than Travis-CI) to confirm that a given
project was indeed NOCI. I would suggest the paper to discuss
this issue, probably in the threats to validity section.

- New metric) I think the paper should address the possibility of correlation between the
'come-in-time' metric (i.e., the new metric) and the existing metrics. For example,
the newly proposed metric is, to some extent, similar to the queue-rank metric.
I wonder whether if these metrics happen to share a correlation. If
that was indeed the case, the explanatory power of the metrics in the
models may change.

Minor comments:

- Ln 363, the paper states that 17 out of 86 projects have acceptable
R-squares. However, it should be 17 out of 87, no?

- It would be nice if the newly collected dataset (i.e., new CI
projects, NOCI projects, and new metrics) could be shared with the community.

References:

Gustavo Sizílio, Daniel Alencar da Costa, and Uirá Kulesza.
An Empirical Study of the Relationship between Continuous Integration and Test Code Evolution. In Proceedings of the 35th IEEE International Conference on Software Maintenance and Evolution (ICSME 2019).

---

## Round 0.2 · accepted · Accept

The revision has carefully addressed the reviewer feedback. The paper sets an example in open science, not just replicating an existing study, but also strengthening its results by means of an alternative experimental set up. The conclusion, that continuous integration does not speed up integrating pull requests but that it does make it possible to handle more such requests, is an important finding. It is with great pleasure that I recommend acceptance of this paper.